

# The Second Curvature Correction for the Straight Segment Approximation of Periodic Vortex Wakes

David H. Wood[1]

[1]Department of Mechanical and Manufacturing Engineering, University of Calgary, Calgary T2N 1N4, AB, Canada.

**Correspondence:** David H. Wood (dhwood@ucalgary.ca)

**Abstract.** The periodic, helical vortex wakes of wind turbines, propellers, and helicopters are often approximated using straight vortex segments which cannot reproduce the binormal velocity associated with the local curvature. This leads to the need for the first curvature correction which is well known and understood. It is less well known that under some circumstances, the binormal velocity determined from straight segments needs a second correction when the periodicity returns the vortex to the proximity of the point at which the velocity is required. This paper analyzes the second correction by modeling the helical far-wake of a wind turbine as an infinite row of equispaced vortex rings of constant radius and circulation. The ring spacing is proportional to the helix pitch. The second correction is required at small vortex pitch, which is typical of the operating conditions of modern large turbines. Then the velocity induced by the periodic wake can greatly exceed the local curvature contribution. The second correction is quadratic in the inverse of the number of segments per ring and linear in the inverse spacing. An approximate expression is developed for the second correction and shown to reduce the errors by an order of magnitude.

## 1 Introduction

It is common for computational models of the wakes of helicopters, propellers, and wind turbines to use straight vortex segments whose position is iterated until they follow the local flow and the vortex is force-free. Solving the Biot-Savart integral gives the induced velocity used in the iteration. Figure 1 shows a representation of a vortex trailing from a two-blade rotor with the straight segment approximation. The labels and symbols on the figure will be defined below. O'Brien et al. (2017) reviewed a range of computational models for wind turbines and Sarmast et al. (2016) describe a recent application of a free-wake vortex model using straight vortex segments.

A well known difficulty of the straight segment approximation is that it does not reproduce the binormal velocity due to the curvature of the vortex line, eg Bhagwat & Leishman (2014), Govindarajan & Leishman (2016), and Kim et al (2016). This leads to the need for the first curvature correction. To assess the errors of the straight segment approximation and develop a correction, Bhagwat & Leishman (2014) used a vortex ring whose binormal (axial) velocity, $U$, is given by the well known Kelvin equation

$$U = \frac{\Gamma}{4\pi}\log\left(\frac{8}{a}\right) - \frac{1}{4} \tag{1}$$

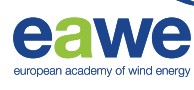



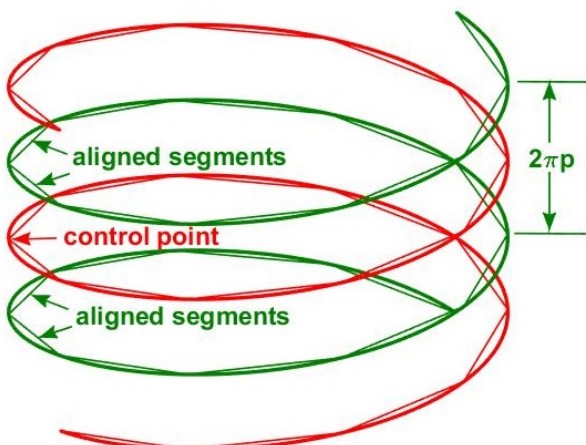

**Figure 1.** Schematic of two turns of constant radius helical vortices with $p = 0.1$ modeling the far-wake of a wind turbine. $N_b = 2$ and the straight segment approximation is shown for $N_s = 10$. The flow is down the page. The "control point" is where the induced velocity is required and the "aligned segments" on either side of the control point are the main contributors to the need for the second curvature correction.

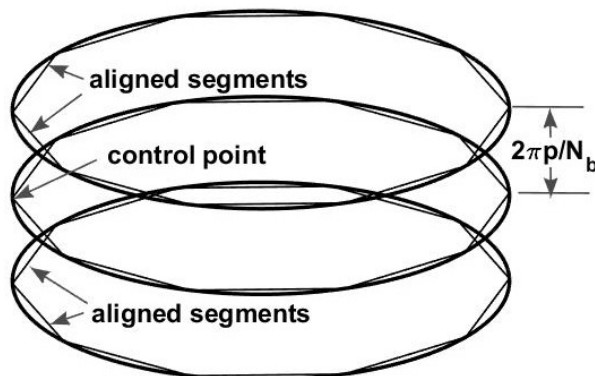

**Figure 2.** Vortex ring representation of the helical wake in Figure 1 and the corresponding vortex segment approximation.

where $\Gamma$ is the circulation of one vortex, and $a$ is the radius of the vortex core, eg Saffman (1992). Figure 2 shows the vortex ring approximation to the helical wake in Figure 1. Note that by Equation (1) $U$ *increases* as $a$ decreases. In this equation, and throughout this paper, all lengths are normalized by the vortex radius (*not the core radius a*), and all velocities by the wind speed. To reproduce Equation (1), the numerical evaluation of the Biot-Savart integral for the ring is "cut-off" by ignoring the contribution from distances smaller than $a$ from the point at which the velocity is required, the "control point" shown in Figures 1 and 2.. It is emphasized that the cut-off is a heuristic; Kelvin's equation (1) is usually derived from impulse considerations or other ways not including the Biot-Savart law. Saffman (1992) documented many factors that alter the vortex velocity from



its Biot-Savart value: these include flow along the vortex axis, differing distributions of swirl etc. Nevertheless, the Biot-Savart prescription is useful and computationally convenient.

Curvature in the wakes of rotors is often associated with vortex periodicity, the "return" of a vortex to the proximity of the control point, which can cause a significant contribution to the binormal velocity. Li & Wood (2002) and Wood (2004) used helical line vortices to analyze straight segment errors for this second effect of curvature, but their work has apparently not been considered in subsequent vortex modeling. Govindarajan & Leishman (2016) claimed that the second curvature correction is unnecessary and difficult to implement. The purpose of this paper is to document the importance of the second correction for wind turbine wakes under some operating conditions and to develop an effective and simple correction.

The paper is organized as follows. The next Section introduces the vortex ring model of the wake. In the following Section, the induced velocity for the periodic component of the wake over a range of vortex spacings is found in terms of its Biot-Savart integral. Section 4 describes the calculation of the induced velocity for the straight segment approximation, determines the second curvature correction, and tests its accuracy. The final Section contains the conclusions.

## 2   The Vortex Ring Model for the Wake

For a point with the same radius as a single vortex ring and distance $z$ from it, the Biot-Savart equation for $U$ in the direction of the wind - the binormal direction - is

$$U = \frac{\Gamma}{4\pi} \int\limits_0^{2\pi} \frac{1 - \cos\theta}{(2 - 2\cos\theta + z^2)^{3/2}} d\theta \qquad (2)$$

where $\theta$ is the vortex angle in cylindrical polar co-ordinates. If $z = 0$, the integral clearly has a logarithmic singularity as $\theta \to 0$. The velocity, $U_{1c}$, requiring the first curvature correction is

$$U_{1c} = U(z = 0) \qquad (3)$$

arising from the only ring containing the control point. The integral in Equation (2) and similar equations will be termed the "influence coefficient" $I$, which has the same relative error characteristics as $U$.

The test case used here to investigate the second correction models the far-wake of a wind turbine as an infinite row of equispaced vortex rings of constant spacing, $s$, radius, and $\Gamma$, extending to infinity on either side of the control point at $z = 0$. A row of rings is easier to analyze than the helical vortices used by Wood & Li (2002) and Wood (2004) but displays the same need at small separation for the second correction. In addition, the discrete nature of the vortex rings helps to localize the correction that is developed in Section 4.

The ring vortex wake is consistent with the "Joukowsky" model of the wake, used by Sarmast et al. (2016); either the bound vorticity of the blades is constant along their span or all the shed vorticity has rolled up into tip and hub vortices before reaching the far-wake. This is clearly a simplification of wind turbine wakes in general, but the linearity of the Biot-Savart law allows more complex wakes to be considered as an assembly of elements such as rings. The velocity associated with the





second correction, $U_{2c}$, is induced by the vortices that do not contain the control point:

$$U_{2c} = \frac{\Gamma}{4\pi} I_{2c} = \frac{\Gamma}{4\pi} \int\limits_{0}^{2\pi} 2 \sum_{j=1}^{\infty} \frac{1 - \cos\theta}{(2 - 2\cos\theta + (js)^2)^{3/2}} d\theta \qquad (4)$$

Equation (4) is not singular as $\theta \to 0$, which is, possibly, the reason why the need for the second correction has not been appreciated. $s$ can be identified with the pitch $p$ of a helical vortex wake and the number of blades, $N_b$, according to $s = 2\pi p / N_b$. The relationship between $p$ and $s$ can be seen by comparing Figures 1 and fig0a.

Testing corrections for the straight segment approximation requires an accurate evaluation of the series in Equation (4) and then an integration in $\theta$. This order is preferred because the integration in $\theta$ of the summand results in incomplete elliptic integrals which are likely to be very difficult to sum. The innocuous looking series in (4), however, does not appear to have a closed form sum. The standard technique for summing infinite series of algebraic functions is via Laplace transforms, eg Wheelon (1954). This would be successful if the exponent in the integrand was 1 instead of $3/2$, but for (4), the method gave a principal value integral that could not be solved in closed form. By the Cauchy integral test for series:

$$\frac{1}{s} - \frac{1}{\sqrt{2 - 2\cos\theta + s^2}} \le I_{2c}(\theta) \le \frac{1}{s} - \frac{1}{\sqrt{2 - 2\cos\theta + s^2}} + \frac{2(1 - \cos\theta)}{(2 - 2\cos\theta + s^2)^{3/2}} \qquad (5)$$

where

$$I_{2c} = \int\limits_{0}^{2\pi} I_{2c}(\theta) d\theta \quad \text{and} \quad I_{2c}(\theta) = \sum_{j=1}^{\infty} \frac{1 - \cos\theta}{(2 - 2\cos\theta + (js)^2)^{3/2}}. \qquad (6)$$

It is easy to show that the average velocity in the direction of the wind at any radius $r < 1$ within the Joukowsky wake is $1 - \Gamma / s$, e.g. Wood (2011), and it is reasonable to assume that the total velocity of the free-wake vortex rings is close to $1 - \Gamma / (2s)$ or the induced velocity, $U \approx \Gamma / (2s)$. Further, the bounds in Equation (5), which both contain $1/s$, cause $U$ to approach the average of the wake and external velocities, *provided the curvature singularity does not contribute significantly to U*. There is, unfortunately, only limited experimental information on $a$ and $U$ for wind turbine wakes to guide the assessment of the relative importance of the first and second velocity fields and their corrections. Figure 3 shows the terms in Equation (5) for $s = 0.2$. This typical value was obtained by the following steps. For modern turbines, $N_b = 3$, and $\lambda \approx 7$ for most of the operating range. For optimal (Betz-Joukowsky) performance, $U = 1/3$, $p \approx 2/(3\lambda)$, where $\lambda$ is the tip speed ratio, and $N_b \Gamma \lambda / \pi = 8/9$, Wood (2011). Thus $\Gamma = 0.133$ and $s \approx 0.2$. The sum in Equation (4) is always zero when $\theta = 0$, but, as $s$ decreases, the bounds in Equation (5) (and hence the sum) tend to $2\pi / s$ over an increasing range of $\theta$. Integrating over $[0, 2\pi]$ then leads to $U \approx U_{2c} \approx \Gamma / (2s)$, showing the potential importance of $U_{2c}$. The integrand of $I_{1c}$ for small $\theta$ is also shown. Its integral and $U_{1c}$ depend on the cut-off, $a$; to match $U = 2/3$ for the conditions in Figure 3, Equation (1) requires $a \sim 10^{-23}$ which does not seem a reasonable value. Thus it is likely that $U \approx U_{2c}$ at the small $p$ and $s$ typical of the operating conditions of modern wind turbines.

The very limited experimental information on the velocity of the vortices in wind turbine wakes are in general agreement with this argument. Xiao et al. (2011) measured the wake of a two-bladed turbine in a wind tunnel at $\lambda = 4.91$ using particle




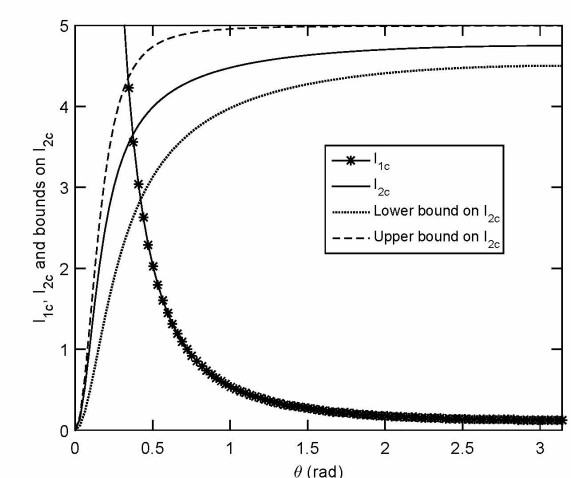

**Figure 3.** The integrands for the two Influence Coefficients for a control point on the vortex ring and $\theta_0 = 0$. The stars show the integrand for $I_{1c}$ from Equation (2) with $z = 0$. The solid line shows $I_{2c}$. The dashed and dotted lines are the bounds on $I_{2c}$, Equation (5) with $s = 0.2$. The sums were evaluated using Equation (4) for 50,000 rings.

image velocimetry. They determined the vortex velocity in the near wake as 10.8 m/s when the wind speed was 12 m/s. Thus $U = (12 - 10.8)/12 = 0.1$ which is lower than the value of $1/3$ that follows from assuming optimal power output. Assuming $U = U_{2c}$ and using the general equation $p = (1-U)/\lambda$, gives $s = 0.576$ or 360 mm for the rotor of radius 625 mm which agrees very well with the value read from their Figure 10. This again implies that $U_{2c} >> U_{1c}$. Since most rotor wakes are helices of

5     some form, it is important to note that the equivalent inverse pitch term dominates $U$ for a helical vortex of sufficiently small $p$, Kuibin & Okulov (1998). There is a further reason to expect $U \approx U_{2c} >> U_{1c}$ for many turbine wakes: $U_{1c}$, but not $U_{2c}$, is associated with the impulse necessary to form a vortex ring. If that impulse and $U_{1c}$ are significant it is unlikely that the wake can be force-free.

## 3    Evaluating the Influence Coefficient for an Infinite Array of Vortex Rings

10    A closed form sum for $U_{2c}$ in Equation (4) could not be obtained so the influence coefficients for a range of $s$ were determined as follows. The Hermite-Hadamard inequality for monotonically decreasing functions that tend to zero at large argument can be used simply to give a tighter bound on $I_{2c}(\theta)$. It is

$$I_{2c}(\theta) = \frac{1}{s} - \frac{1}{\sqrt{2 - 2\cos\theta + s^2}} + \frac{1 - \cos\theta}{(2 - 2\cos\theta + s^2)^{3/2}} + \delta(\theta) \qquad (7)$$

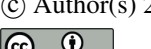



where the difference, $\delta(\theta)$, is always positive positive but must be determined numerically. The integral of the other terms on the right side of (7) can be found exactly:

$$\int_0^{2\pi} \left( \frac{1}{s} - \frac{1}{\sqrt{2-2\cos\theta+s^2}} + \frac{1-\cos\theta}{(2-2\cos\theta+s^2)^{3/2}} \right) d\theta = \frac{2\pi}{s} - \frac{2sE(-4/s^2)}{s^2+4} - \frac{2K(-4/s^2)}{s} \tag{8}$$

where $E$ and $K$ are the complete elliptic integrals in standard notation. The difference, $\delta$, the integral of $\delta(\theta)$ over $[0, 2\pi]$, was

evaluated using 2,000 increments of $\theta$ and number of rings, $N_r = 50,000$. This value was chosen using

$$\int_0^{2\pi} 2 \sum_{j=N_r}^{\infty} \frac{1-\cos\theta}{(2-2\cos\theta+(js)^2)^{3/2}} d\theta \to R(N_r) = \frac{4\pi\left(\zeta(3) - H_{N_r}^{(3)}\right)}{s^3} \tag{9}$$

for large $N_r$, where $\zeta(.)$ is the zeta function, and $H(.)$ is the harmonic number in standard notation. In later use of this result, $R(j)$ will be called the "remainder". Using $2\pi/s$ as an estimate for the integral in (4), and using *Mathematica* to evaluate $H_3(N_r)$, gave the relative error in truncating the sum at $N_r = 50,000$ as $2/s^2 \times 10^{-10} = 2 \times 10^{-8}$ for the smallest value of $s$

considered here, $s = 0.1$. To the number of decimal places used in Table 1, truncation does not alter the integral of the terms in Equation (7) over $[0, 2\pi]$. For every calculation up to Section 5, $N_r = 50,000$. The $\theta$−integral of the sum in Equation (7) was found using the Matlab quadrature routine *integral* for an absolute tolerance of $10^{-8}$. All integrands are symmetric about $\theta = \pi$ and so were obtained over $[0, \pi]$. Table 1 also shows the approximate $\lambda$ for a Betz-Joukowsky optimal rotor. $\delta$ is small: combining Equations (6) and (7) leads to

$$0 \le \delta \le \frac{2K(-4/s^2)}{s} - \frac{2sE(-4/s^2)}{s^2+4} \tag{10}$$

which is satisfied by all $\delta$ in Table 1. $\delta$ is less than 4% of $I_c$ for the worst case of $s = 0.8$; Equation (8) is an increasingly good approximation to the sum and the influence coefficient approaches $2\pi/s$ as $s$ decreases and $\lambda$ increases.

**Table 1.** Values of the Influence Coefficient for varying $s$ with $\theta_0 = 0$.

| $s$ | Approx. $\lambda$ | Equation (8) | $\delta$ | $I_{2c}$ | $2\pi/s$ |
|------|------|------|------|------|------|
| 0.10 | 14 | 57.448345 | 0.163716 | 57.612061 | 62.831853 |
| 0.20 | 7 | 26.722916 | 0166712 | 26.889628 | 31.415927 |
| 0.40 | 3.5 | 11.703237 | 0.172962 | 11.876199 | 15.707963 |
| 0.80 | 1.75 | 4.546484 | 0.174676 | 4.721160 | 7.853982 |

The values of $I_{2c}$ in Table 1 will be compared to the values from the straight segment approximation.

## 4   Straight Segment Approximation of Vortex Rings and Its Accuracy

Each of the rings not containing the control point was approximated by an even number, $N_s$, of straight segments. The $i$th segment of each ring started at $\theta = 2\pi i/N_s + \theta_0$ when measured from the control point and finished at $2\pi(i+1)/N_s + \theta_0$. $\theta_0$



is the angular displacement between the control point and the start of the first ($i = 0$) segment. A straightforward application of Equation (10.115) of Katz & Plotkin (2001) gives $I_{2c}(i, j)$, the contribution of the $i-$th segment on the two $j-$th vortex rings to $U_{2c}$, as

$$U_{2c}(i,j) = \frac{\Gamma}{4\pi} \sum_{j=1}^{N_r} \sum_{i=1}^{N_s} I_{2c}(i,j) \quad \text{where} \quad I_{2c}(i,j) = \frac{A_i B_{i,j}}{C_{i,j}}, \tag{11}$$

5    and

$$A_i = 8 \sin\left(\frac{\pi i}{N_s} + \frac{\theta_0}{2}\right) \sin\left(\frac{\pi(i+1)}{N_s} + \frac{\theta_0}{2}\right), \tag{12}$$

$$B_{i,j} = \sin\frac{\pi}{N_s}\left(\frac{1}{b_{i+1,j}} + \frac{1}{b_{i,j}}\right) + \sin\left(\frac{(2i+1)\pi}{N_s} + \theta_0\right)\left(\frac{1}{b_{i+1,j}} - \frac{1}{b_{i,j}}\right), \tag{13}$$

$$C_{i,j} = \cos\frac{2\pi}{N_s} + b_{i,j}^2 + b_{i+1,j}^2 + \cos\left(\frac{2\pi(2i+1)}{N_s} + 2\theta_0\right) - 2, \tag{14}$$

and

10    $$b_{i,j} = \sqrt{2 - 2\cos\left(\frac{2\pi i}{N_s} + \theta_0\right) + (js)^2}. \tag{15}$$

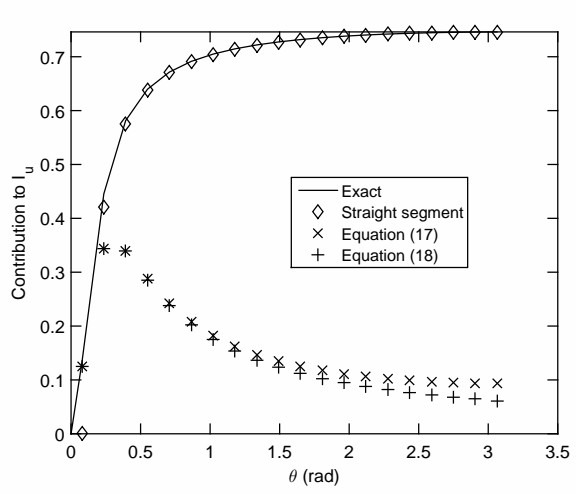

**Figure 4.** Angular contribution of straight segments to the Influence Coefficient for $z = 0.2$, $N_s = 40$, and $\theta_0 = 0$. The value of $\theta$ is the midpoint of the segment. The solid line shows the numerical solution of the exact integral. $\times$, Equation (17); $\circ$, Equation (17) for the first two segments from $\theta = 0$.

For the first calculations, the junction of the first and $N_s$th segment was aligned with the control point so $\theta_0 = 0$. The influence coefficients calculated from Equations (11-14) are compared to the results from Table 1 in Figure 5 in terms of the relative error using the integral over $[0, \pi]$ as the denominator since the integrand must be symmetric about $\theta = \pi$. The error is defined as the



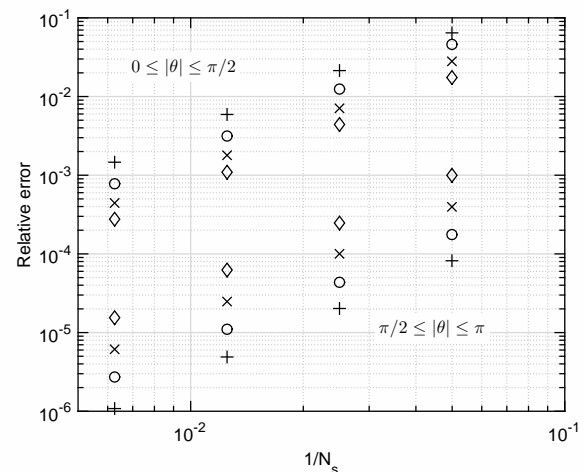

**Figure 5.** Relative error of the straight segment approximation for the conditions in Table 1 ($\theta_0 = 0$). $+, s = 0.1; \circ, s = 0.2; \times, s = 0.4; \Diamond, s = 0.8$. The top group of symbols show the errors for the first half of the rings that contain the aligned segments, $0 \leq |\theta| \leq \pi/2$, and bottom group covers $\pi/2 \leq |\theta| \leq \pi$. $N_s = 20, 40, 80, and 160$.

exact integral minus the straight segment approximation so the latter is aways an under-estimate. Two separate ranges of $\theta$ are considered: for $0 \leq |\theta| \leq \pi/2$ the error is one to two orders of magnitude higher than for $\pi/2 \leq |\theta| \leq \pi$ and the errors in the first range increase with decreasing $s$ whereas in the second range they decrease. Figure 4 shows the reason. For this test case, no aligned segment contributes to $U_{2c}$ as the Biot-Savart velocity must lie in the plane containing the segment and the control

point. Otherwise, both errors scale as $1/N_s^2$, as was found in the helix simulations of Wood (2004). As $s$ decreases, however, the aligned segment error increases proportionally to $1/s$, but the error for the remaining range of $\theta$ decreases at constant $N_s$.

Figure 4 shows the angular contribution to the influence coefficients for $s = 0.2$ and $N_s = 40$. The value of $\theta$ used for plotting is the midpoint of each segment. The solid line shows the exact integral from $\theta_i$ to $\theta_{i+1}$. As was found by Wood & Li (2002) and Wood (2004), the errors are localized near $\theta = 0$. A correction for the error for the aligned segments can be developed

from the small-$\theta$ expansion of the series in Equation (4):

$$I_{2c}(\theta) = 2 \sum_{j=1}^{\infty} I_{2c}(\theta, j) = \sum_{j=1}^{\infty} \frac{2 - 2\cos\theta}{(2 - 2\cos\theta + (js)^2)^{3/2}} \rightarrow \frac{\zeta(3)\theta^2}{s^3} \quad \text{as} \quad \theta/s \rightarrow 0 \tag{16}$$

and $\zeta(3) = 1.2026$. Equation (16) has two important implications. First, the best possible error for periodic straight segments scales as $(N_s s)^{-3}$ but it is likely that an unrealistically high value of $N_s$ would be required to achieve this. Second, $1/\zeta(3)$ or over $80\%$ of the correction to $U_{2c}$ is due to the two rings ($j = 1$) on either side of the control point. A general form of the

correction, therefore, can be based on the returned vortex on either side of the control point. Since the distance from the control point to the vortex segments must be calculated in a free-wake simulation, it should not be difficult to determine the proximity





in terms of $\theta/z$ and apply a correction. A more general correction is $\Delta(\theta_s)$, where $\theta_s = 2\pi/N_s$ is obtained by integrating in $\theta$ only for $j = 1$ and then using $\zeta(3)$ to correct approximately for the remaining rings. The result is

$$\Delta(\theta_s) \approx 2\zeta(3)\Big[\frac{F\left(\theta_s/2, -4/s^2\right)}{s} - \frac{sE\left(\theta_s/2, -4/s^2\right)}{s^2+4} - \frac{2\sin\theta_s}{(s^2+4)\sqrt{2-2\cos\theta_s+s^2}}\Big] \tag{17}$$

where $E$ and $F$ are the incomplete elliptic integrals. Equation (17) is shown in Figure 3 to give a better estimate for the aligned

segments. Unfortunately, approximate expressions for incomplete elliptic integrals with negative modulii - $-4/s^2$ in Equation (17) - are not available. An alternative, simpler correction than (17) can be found by using $2 - 2\cos\theta \sim \theta^2$ for small $\theta$ to give:

$$\Delta(\theta_s) \approx 2\zeta(3)\left[\log\left(\frac{\theta_s + \sqrt{\theta_s^2 + s^2}}{s}\right) - \frac{\theta_s}{\sqrt{\theta_s^2 + s^2}}\right] \tag{18}$$

which gives almost the same correction for the aligned segments, Figure 4. These results for the application of Equations (17) and (18) to the aligned segments are similar at the other values of $s$ as well, but are not shown in the interests of brevity. It is

noted that the correction developed here is simple in the sense that the vortex curvature is known *a priori*. As pointed out by Govindarajan & Leishman (2016), however, and shown by the analysis of Kim et al. (2016), the modeling of three-dimensional wakes of varying geometry can be considerably more complex.

One of these complexities is that the control point may not align with the junction of segments on (in this case) adjacent rings. The effect of this can be investigated by using non-zero $\theta_0$ in Equations (12)-(15). The results are shown in Figure 6

for $20 \le N_s \le 160$ and $0 \le \theta_0 \le \pi/N_s$ and $s = 0.1$. As was found for other values of $s$, there is remarkably little variation in the error with $\theta_0$ except for the lowest $N_s$, suggesting that the correction derived above for the aligned case ($\theta_0 = 0$) is also applicable to other values. This is not an immediately obvious result from Equations (12)-(15). For $\theta_0 = \pi/N_s$ and $\theta = 2\pi/N_s$:

$$I_{2c}(\theta, 1) = \frac{8\sin^2(\theta/2)\sin\theta}{(3 - 4\cos\theta + \cos2\theta + 2s^2)\sqrt{2 - 2\cos\theta + s^2}} \rightarrow \left(\frac{\theta}{s}\right)^3 \text{ as } \theta/s \to 0 \tag{19}$$

which suggests a difference from the case when $\theta_0 = 0$,

## 5    Using a Finite Array of Rings to determine the Influence Coefficient

The second curvature error was shown in the last section to be caused largely by aligned segments on the rings either side of the control point. For increasing $\theta$, $I_{2c}(\theta)$ becomes dominated by rings at larger distance from the control point. This is shown in Figure 7 which implies that $N_r$ either must be large to ensure an accurate determination of $I_{2c}$ or a suitable "remainder"

term be used. This allows an approximate determination of the influence coefficient for the case where $\theta_0 = 0$ according to

$$I_{2c} \approx 2\Delta(\theta_s) + \sum_{j=1}^{N_r}\sum_{i=1}^{N_s} I_{2c}(i,j) + R(N_r) \tag{20}$$

where one possibility for the remainder, $R(N_r)$ is given by Equation (9). The terms in Equation (20) are listed in Table 2 for $s = 0.2$. A significant number of vortex rings, $N_r$, or, equivalently a large streamwise distance is needed to make the

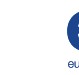



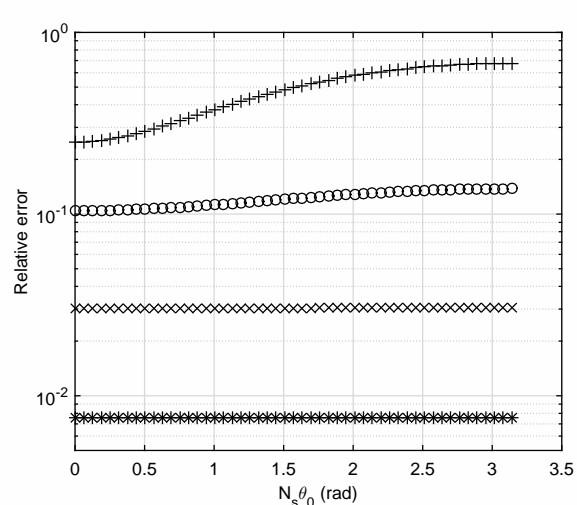

**Figure 6.** Variation of relative error with $\theta_0$ for $s = 0.1$. $N_s = 20, +; N_s = 40, \circ; N_s = 80, \times; N_s = 160, \star;$

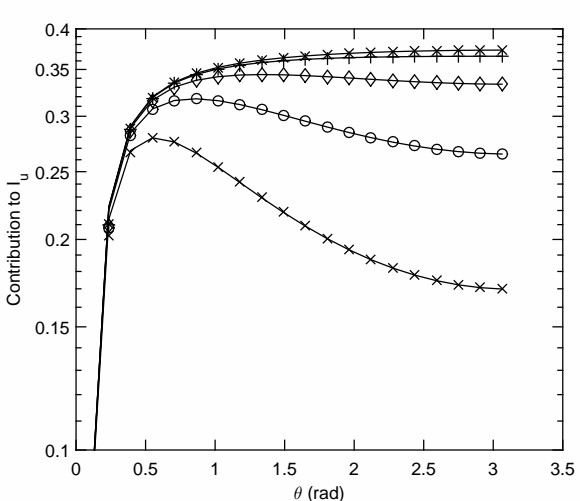

**Figure 7.** Variation of the straight segment approximation to $I_{2c}(\theta)$ for $\theta_0 = 0$ and $N_r =, 5, \times; 10, \circ; 20, \diamond; +, 50; \times, 50,000$. $N_s = 40$, $s = 0.2$.

remainder, $R(N_r)$, accurate. Typically, $N_r \geq 20$ for this $s$, and then $R(N_r)$ is comparable to $\Delta(\theta_s)$. For $N_s = 20$, for example, after applying an accurate remainder, the second curvature correction changes the relative error from 3.4% to less than 0.2% which is a reduction by two orders of magnitude.





**Table 2.** Terms in Equation (20) for $s = 0.2$ and $\theta_0 = 0$. Exact value of $I_{2c} = 26.889628$. The error is the relative error.

| $N_r$ | $N_s$ | $2\Delta(\theta_s)$ | $\sum_{j=1}^{N_r} \sum_{i=1}^{N_s} I_{2c}(i,j)$ | $R(N_r)$ | Approx. $I_{2c}$ | Error |
|---|---|---|---|---|---|---|
| 5 | 20 | 0.937219 | 15.510093 | 25.752995 | 42.200307 | -0.569 |
| 10 | 20 | 0.937219 | 20.873245 | 7.107724 | 28.918187 | -0.0754 |
| 20 | 20 | 0.937219 | 24.023349 | 1.867772 | 26.828341 | 0.0023 |
| 50 | 20 | 0.937219 | 25.353797 | 0.307939 | 26.598955 | 0.0108 |
| 50,000 | 20 | 0.937219 | 25.650226 | 0.0 | 26.587445 | 0.0112 |
| 5 | 40 | 0.248964 | 16.294055 | 25.752995 | 42.296015 | -0.573 |
| 10 | 40 | 0.248964 | 21.719497 | 7.107724 | 29.076185 | -0.0813 |
| 20 | 40 | 0.248964 | 24.904960 | 1.867772 | 27.021697 | -0.0049 |
| 50 | 40 | 0.248964 | 26.251313 | 0.307939 | 26.808216 | 0.0030 |
| 50,000 | 40 | 0.248964 | 26.551412 | 0.0 | 26.800037 | 0.0033 |

## 6 Conclusions

The widely used straight segment approximation for approximating the curved and periodic vortex wakes of wind turbines, propellers, and helicopters can have two errors associated with the wake curvature. The first is the well-known error in reproducing the locally-induced binormal velocity. This is usually accommodated by a cut-off in the Biot-Savart determination of
5 the vortex velocity using Equations (2) and (3) at a distance comparable with the radius of the vortex core. The second, less well-known error, is the subject of this paper. It arises from the alignment of the segments of the periodic vortex returning to the proximity of the point at which the velocity is being determined.

By modeling the far-wake of a wind turbine as an infinite row of equispaced vortex rings, two important results were obtained. First, it was shown that the velocity associated with the second error dominates at the small spacings typical of
10 modern wind turbine operation. The available experimental evidence on wake structure is consistent with this finding. Second, it is shown that the second error is quadratic in the number of segments per revolution and inversely proportional to the spacing of the rings which is proportional to the pitch of a more realistic, but more difficult, helical wake. The model to investigate the second correction is artificial in that a single, infinite row of vortex rings of constant spacing, radius, and circulation is not generally applicable to wind turbine wakes. Nevertheless the model demonstrated the importance of the rings adjacent to the
15 control point at which the velocity is being calculated. For the ring model, those adjacent rings contribute over 80% of the correction developed. The correction is needed because the straight segment approximation

It was also shown that the best behaviour possible for the second error is cubic in the product of the number of segments per revolution and the vortex spacing. It is likely, however, that larger numbers of vortex segments would be needed to achieve this error than are used in practice . This result was generalized to develop a second correction that improves the computed induced
velocity by nearly one order of magnitude.



*Competing interests.* The author claims no competing interests.

*Code availability.* The Matlab codes used in this study are available from the author.

*Acknowledgements.* This work is part of a research project on wind turbine aerodynamics funded by the NSERC Discovery Program.





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
