# Peer review of "The Second Curvature Correction for the Straight Segment Approximation of Periodic Vortex Wakes"

_Wind Energy Science, 2018_

## Referee Comment (RC1) · W. Xiaodong (Referee) · 15 Mar 2018

The simulation of vortex filaments movement is important for the free-vortex wake method which having been widely used for aerodynamic predictions of wind turbine and helicopter rotors. This paper investigated the second curvature error of induction velocity of curved vortex filaments and presented a model for induction velocity correction of vortex ring model. This paper shows high quality scientific contribution to the free-vortex method improvements and engineering applications.

Detailed comments 1. The induction velocities of blades are mostly contributed by the induction of near wake, which was expanded and helical lines. How does the second

curvature error affect the blade induction velocities and loads?

2. The free-vortex wake is based on potential flow assumption. Theoretically, the vortex wake was convected to downwind infinite distance. However, the concentrated tip vortex only existed a short distance and was dissipated in far wake. Author tested the correction model for far wake models based on vortex theory. What is the engineering significance of far wake modelling using free-vortex wake?

3. In the introduction, the author mentioned that "The purpose of this paper is to document the importance of the second correction for wind turbine wakes under some operating conditions". In my understanding, this operating condition is normal power production condition for wind turbine. Particularly, the modern large diameter wind turbine designed for the low wind speed operates at higher tip speed ratio (10-12) usually. It means the space of vortex rings, s, is a relative small value compared to rotor diameter. I suggest the author could add the s value for this type of wind turbine.

4. This paper will be highly interesting to engineering application if the author could explain more how to implement the second correction model for helical expanded non-periodic wake.

---

## Referee Comment (RC2) · J. Saverin (Referee) · 16 Mar 2018

General Comments:

This paper discusses the second curvature correction applied to the treatment of a wind turbine wake with vortex filaments. From a purely theoretical point of view, the results are quite interesting and the methods used appear valid. It is shown which terms (or vortex rings, and their segments) contribute the most to this correction.

Although as a theoretical treatment of an idealized case, the work is quite facinating, it is to the author's opinion unlikely that the result will be applied to numerical modelling

of a wind turbine wake. The main advantage of using vortex filaments to treat the wake is that they are treated as material elements and convect under the local velocity field, so they allow the changes to the induced velocity due to the wake deformation to be treated. This method applies only to aligned segments, which hence represent a prescribed wake. In the case of the deformed wake, the self-induction corrction then appears to be more relevant.

Specific Comments:

- The expression from Equation 9 seemingly appears from nowhere. Is this an analytical result? - Regarding Equations 11-15. Is it really necessary to show these equations? They are extracted from a relatively simple equation, and were evaluated numerically anyway. Does presenting them in this (seemingly more complicated form) serve any purpose? - Page 10: Is this not a reduction by only "one" order of magnitude?

Technical Comments:

- Page 4: "Figures 1 and fig0a" - Page 5: Caption: Influence Coefficients capitalized - Page 6: "positive positive" - Page 7: Caption, Figure 4: "×, Equation (17);o Equation (17)" - Why did the author refer first to Figure 5, however Figure 4 appears first in the document. Perhaps better to reorder. - Page 8: Figure 5: Caption: Ns = ....'and' (italics) 160. - Conclusion: The correction is needed because the straight segment approximation....?

---

## Editor Comment (EC1) · A. Bianchini (Editor) · 19 Mar 2018

Dear author, your work received some interesting comments and suggestions for improvement. You are kindly requested to revise the paper according to all of them and upload a new version. Best regards

---

## Author Comment (AC1) · 13 Apr 2018

Reviewer 1

I thank Dr Xiaodong for his comments.  I will respond in order:

1. I do not know how errors caused by neglecting the second correction will affect the accuracy of determining the flow over the blades.  I suspect that this important question can only be answered by a thorough study using a high quality free-wake method with and without the correction.  I have started collaboration with the developer of such a method and hope to be able to answer the question in the near future.

2. The reviewer is correct in asserting that, in practice, the concentrated tip vortices are dissipated or destroyed within a short distance downwind of an operating rotor in the atmospheric boundary layer. Both the first and second curvature corrections are needed, in principle, whenever there is vortex curvature and vortex return, which is the case immediately behind the rotor as in the far-wake.  The latter is, however, easy to analyze and provide accurate corrections which, hopefully, can be used in any situation.

3. The statement was made because the criticism of the need for the second correction was made by researchers working on helicopter wake modeling where periodic wakes are less common.  I wanted to emphasize that the second correction is required only when proximity of the returning vortex brings the aligned segments close to the control point. I take the reference to 2-bladed turbines to be to the MingYang SCD series (http://www.mywind.com.cn/upfile/File/2012/SCD%20Series%20Wind%20Turbine%20Generator.pdf). The product brochure gives the rated speed as 11 m/s and the tip speed as 89.6 m/s.  This gives the tip speed ratio, TSR = 8.1,  This is likely to underestimate the TSR in region 2 but I could find no further documentation on TSR.  This does not seem a large enough increase on the values quoted in the paper to be worth including.  If Dr Xiaodong has further information I would be grateful for it.

4. Wake expansion is likely to reduce the importance of the second correction in comparison to a constant diameter wake of the same pitch. Wood (2004) investigated the second correction for expanding helical wakes but used a more complex and less accessible methodology. The intention with the current paper was to simply introduce the second correction.

Reviewer 2

I thank Dr Saverin for his comments. I disagree with the interpretation that aligned segments occur only in a fixed wake method.  Free wake methods (FWMs) are often in need of the correction because, as the reviewer noted, they require accurate estimates of the induced velocity to align the vortex segments with the local flow.  A FWM should be able to model a constant diameter and constant pitch far-wake as this is the easiest possible case and the one most in need of correction at small p or s, otherwise the vortex velocity cannot be the average of the wind speed and the velocity within the wake which depends on p or s.  The paper does not claim the second correction is universal.  As noted in response to Reviewer #1, and as indicated by the statement of the aim of the paper in lines 7 and 8 of page 3 and in the conclusion, it is likely that most helicopter wakes, for example, are unlikely to require the second correction.

In response to the specific comments:

1. The right side of Equation (9) is the closed form of the integral from Mathematica. The text has been altered to make this clear.

2. Equations (11-15) for the straight segment approximation are needed to establish the accuracy of the analysis. A mistake in them would invalidate the correction proposed in the paper so I deemed it essential to include the equations.

3. The reviewer is correct: "nearly" should not appear in discussing orders of magnitude and has been removed.

I thank the reviewer for the close reading of the manuscript. The technical comments have been addressed in the revised manuscript.

One additional correction was made. Since the paper was submitted, I was made aware that there is a transformation of incomplete elliptic integrals with negative moduli to positive moduli. Consequently I removed the sentence between Equations (17) and (18) about the difficulty in approximating these functions has been removed. I have also taken the opportunity to make some additional small changes to the text to improve clarity.